# Synthesis and Physico-Chemical Analysis of Dextran from Maltodextrin via pH Controlled Fermentation by *Gluconobacter oxydans*

**DOI:** 10.3390/foods14010085

**Published:** 2025-01-01

**Authors:** Seung-Min Baek, Bo-Ram Park, Legesse Shiferaw Chewaka, Yun-Sang So, Ji-Hye Jung, Seul Lee, Ji Young Park

**Affiliations:** 1Fermentation Research Department, National Institute of Agricultural Science, RDA, Jeonju 54875, Republic of Korea; 2Department of Food Science & Biotechnology and Carbohydrate Bioproduct Research Center, Sejong University, Seoul 05006, Republic of Korea

**Keywords:** α-1,6 glucan, bioconversion, dextran, EPS, *Gluconobacter oxydans*

## Abstract

Dextran is an exopolysaccharide (EPS) with multifunctional applications in the food and pharmaceutical industries, primarily synthesized from *Leuconostoc mesenteroides*. Dextran can be produced from dextrin through *Gluconobacter oxydans* fermentation, utilizing its dextran dextrinase activity. This study examined how jar fermentor conditions impact the growth and enzyme activity of *G. oxydans*, with a focus on the effects of pH on dextran synthesis via bioconversion (without pH control, pH 4.5, and pH 5.0; Jp-UC, Jp-4.5, and Jp-5.0). After 72 h, the cell density (O.D. at 600 nm) was 7.2 for Jp-4.5, 6.5 for Jp-5.0, and 3.7 for Jp-UC. Flow property analysis, indicating dextran production, showed that Jp-4.5 had the highest viscosity (30.99 mPa·s). ^1^H-NMR analysis confirmed the formation of α-1,6 glycosidic bonds in bioconversion products, with bond ratios ranging from ~1:0.17 to ~1:2.84. The distribution of molecular weights varied from 1.3 × 10^3^ Da to 5.1 × 10^4^ Da depending on pH. The hydrolysis rates to glucose differed with pH, with the slowest rate at pH 4.5 (53.96%). These results suggest that the production of dextran by *G. oxydans* is significantly influenced by the pH conditions. This dextran could function as a slowly digestible carbohydrate, aiding in postprandial glycemic regulation and mitigating chronic metabolic diseases like diabetes.

## 1. Introduction

Microorganisms produce exopolysaccharides (EPS) to adapt to their environment [1,2]. In general, pathogenic microorganisms use EPS to form biofilms to attach extracellular cells and protect against environmental factors [3,4,5]. Acetic acid bacteria (AAB), such as *Gluconobacter oxydans*, can produce β-glucan structures like bacterial cellulose (BC) or α-glucan structures like dextran under specific conditions [6,7]. BC is a water-insoluble polymer composed of β-1,4 glucan chains, which differentiate it from dextran, a water-soluble polymer primarily composed of α-1,6 glucosidic linkages with variable amounts of α-1,4-, α-1,3-, and α-1,2 linkages [6,8,9,10]. The production of these biopolymers depends on culture conditions such as pH, medium composition, and enzymatic activity.

Dextran is homopolymers of D-glucopyranosyl residues that mainly synthesized from sucrose by *Leuconostoc* spp. via transglucosylation by dextransucrases (DSases; E.C. 2.4.1.5) that produce dextran containing a high percentage of consecutive α-1,6 linkages and a low percentage of α-1,3 linkages [11,12]. In 1949, Hehre and Hamilton [13] showed that polysaccharides from Acetobacter capsulatum (i.e., *Gluconobacter oxydans*) could produce ropy beer, similar to polysaccharides from *Leuconostoc mesenteroides*. Additionally, Hehre [14] and Yamamoto et al. [15] reported that dextran dextrinase (DDase; E.C. 2.4.1.2), derived from *G. oxydans*, could convert dextran from maltodextrin (MD) in contrast to DSase. The structure of dextran synthesized by *G. oxydans* DDase harbored α-1,4 branches and α-1,4 linkages in α-1,6 glucosyl linear chains [16,17]. In particular, DDase has an advantage over DSase in the dextran production process because DDase does not produce monosaccharide by-products such as fructose [18,19].

Dextran is a biopolymer isolated from microorganisms that use dextran to form parts of their viscous extracellular matrix [20]. It is industrially useful polymers that do not exist in nature and have demonstrated multifunctional applications in the biochemical (e.g., molecular sieves for chromatography) and pharmaceutical (e.g., blood plasma substitute) industries [9,18]. Moreover, these microbial polysaccharides can exert distinct functions depending on their molecular weight (MW), the targeted saccharide substrate, the position of linkages, and the presence or absence of branched bonds [21]. Given their unique structural characteristics, dextrans from *Gluconobacter* spp. have been used in the food industry as sources of dietary fiber, cryostabilizers, fat substitutes, and low-calorie bulking agents for sweeteners [20]. Furthermore, dextrans have been applied broadly in other industries including as components for cosmetics [22], dietary fiber due to its low digestibility by intestinal enzymes [17], in high-viscosity gums, and as food additives [23].

Numerous studies have reported the production and characteristics of dextran derived from the heterogeneous expression of *G. oxydans* DDase [12,15,16,24,25]. However, the use of this enzyme as a food ingredient requires approval from government food agencies. Additionally, the production of biomaterials via the fermentation of AAB demonstrating enzyme activity might be affected by various factors including medium composition, substrate concentration, and culture temperature. Furthermore, previous studies have reported on the increased productivity of *G. oxydans* in bioreactors due to their lower biomass [26,27]. Nevertheless, there is a knowledge gap associated with the establishment of the optimal culture conditions for efficient dextran production, especially with regard to the optimal pH employed during semi-continuous fermenter operations.

Therefore, in this study, we analyzed the rheological characteristics of the culture media and physiochemical characteristics of *G. oxydans* dextran synthesized in dextrin media at different pH values. These results provide fundamental information that can be applied to generate functional carbohydrates as food raw materials via the bioconversion process.

## 2. Materials and Methods

### 2.1. Cell Strains and Preparation of Stock Cultures

*G. oxydans* KACC 19,357 (ATCC 11894) was obtained from the Korean Agricultural Culture Collection (KACC, Suwon, Korea). The bacteria were grown at 30 °C in acetic acid bacterium (AAB) medium containing 5 g bactopeptone, 5 g yeast extract, 5 g D-glucose, and 1 g MgSO_4_∙7H_2_O per liter at a pH range of 6.6 to 7.0. The organisms were initially cultured on AAB agar medium at 30 °C for 72 h and then inoculated into liquid medium at 30 °C for 24 h. At the end of the incubation time, stock cultures were prepared by dispensing 1 mL of the inoculum into sterile 2 mL cryovials containing 0.5 mL glycerol. The resulting suspensions were mixed and stored at −70 °C.

### 2.2. Cultivation of the Jar Fermenter

Dextran production by *G. oxydans* was conducted in a 5.0-L jar fermenter (Kobiotech, Incheon, Korea). Jar fermenter cultures were performed as follows. The strain was precultivated for 20 h to an optical density at 600 nm (OD_600_) of 0.5 in a 250-mL flask containing 60 mL AAB medium at 30 °C with shaking at 200 rpm. The media from the seed culture were then transferred to media in the 2.5-L jar fermenter media, the composition of which has been previously described [28,29,30]. The culture conditions including dextrin composition, injection time, and shaking speed were established and slightly modified in several preliminary experiments [31]. The final culture medium comprised 1.2 L of media comprising 6 g yeast extract, 0.6 g KH_2_PO_4_, 0.6 g K_2_HPO_4_, 6 g D-glucose, 1.2 g MgSO_4_∙7H_2_O, 60 g MD [dextrose-equivalent; Serimfood, Buchoen, Korea), and 2% (*v*/*v*) glycerol. Additionally, we manually added substrate solution (60 g MD, 2 g MgSO_4_∙7H_2_O, and 100 mL distilled water) at 50-mL increments at 12 h and 24 h of culture. The pH of the culture was controlled by the automated addition of 10% (*v*/*v*) NaOH and 50% (*v*/*v*) glycerol solution. Foam content was automatically controlled by the addition of 0.05% (*v*/*v*) antifoam solution (Antifoam 204; Sigma-Aldrich, St. Louis, MO, USA). A pH of 4.5 is considered optimal for DDase activity, and a pH of 5.0 is considered optimal for cell-mass proliferation and DDase stability following isolation from acetic acid bacteria [15,32]. According to previous results showing that pH affects the growth of *G. oxydans* [33], we compared the culture characteristics of *G. oxydans* under different pH values by employing three cultures with different pH status (Jp-UC, jar fermentor pH uncontrolled; Jp-4.5, pH controlled to 4.5; and Jp-5.0, pH controlled to 5.0) (Table 1). The OD_600_ of the cultures was determined using a 1 mL cuvette in a UV–Vis spectrophotometer (Cary 3500 UV–Vis; Agilent Technologies, Santa Clara, CA, USA). Samples were appropriately diluted in order to maintain the OD_600_ value between 0.2 and 0.8.

### 2.3. Analysis of Rheological Properties

We performed rheological measurements using a modified method from Im et al. [34] to describe the flow behaviors of the culture medium while using 10% and 20% dextran solution (dextran 40; Tokyo Chemical Industry, Tokyo, Japan) as positive controls. The flow sweep test was conducted using a rheometer (Discovery HR-1; TA Instruments, New Castle, DE, USA) interfaced with TRIOS software (V5.6, TA Instruments). All experiments were performed using Peltier plate geometry (diameter: 40 mm) at 25 °C. The viscosity was measured as a function of shear rate in the range of 0.1 s^−1^ to 300 s^−1^, with apparent viscosity expressed at a shear rate of 100 s^−1^.

### 2.4. Dextran Isolation

The fermented media were centrifuged in 7119× *g* for 10 min at 4 °C to remove the cells, after which dextran was precipitated from the culture supernatant following the addition of nine volumes of ice-cold absolute ethanol and centrifuged after incubation at −80 °C for 2 h. The precipitate was freeze-dried, with the powder form used for analysis (Appendix A).

### 2.5. Gel Permeation Chromatography (GPC) Analysis

Gel permeation chromatography (GPC) was used to analyze the molecular weight distribution of the polysaccharides produced as described before [35]. The culture solution was treated with 90% (*v*/*v*) cold ethanol to precipitate the polysaccharides, which were then recovered and freeze-dried. The sample (10 mg/mL) was dissolved in distilled water, filtered through a 0.22 μm syringe filter (13 mm, 0.22 μm, NYLON, Thermo Fisher Scientific Co., Rockwood, TN, USA), and used for analysis. An HPLC system (Agilent Technologies 1260 Infinity, Waldbronn, Germany) equipped with a TSK gel G3000PW column (7.8 mm × 30 cm, Tosoh, Tokyo, Japan) was used for separation. The column was maintained at 40 °C with a sample injection volume of 10 μL and distilled water as the solvent. Molecular weight was determined by elution time, using the glucose, maltose, and pullulan standards from Sigma-Aldrich as reference substances.

### 2.6. NMR Analysis

The ratio of α-1,4 and α-1,6 glycosidic bonds in high molecular weight polysaccharides produced under different culture conditions was analyzed using ^1^H-NMR spectroscopy (500 MHz FT-NMR, JEOL, Tokyo, Japan). Polysaccharides were recovered by ethanol precipitation from the culture supernatant, and the freeze-dried sample (20 mg/mL) was dissolved in deuterium oxide (D_2_O). The solution was incubated at 45 °C for 20 min before performing the ^1^H-NMR analysis [36]. The parameters were as follows: spectrometer frequency = 500.16 MHz; spectral width = 9384.4 Hz; acquisition time = 3.5 s; relaxation delay = 20 s; number of scans = 16; 90° pulse width = 7.6 μs; pulse angle = 45°; digital resolution = 0.29 Hz.

### 2.7. Hydrolysis Properties of Dextran by Mammalian Mucosal α-Glucosidase

To investigate the digestibility characteristics of the dextran, the glucose content released by the action of the digestive enzyme RIAP (rat intestinal acetone powder), a mammalian α-glucosidase, was measured using a modified method from Um et al. [37]. A freeze-dried polymer sample, recovered by ethanol precipitation from the culture medium, was dissolved at a concentration of 0.1% (*w*/*v*) in 100 mM phosphate buffer (pH 6.0) containing 6.7 mM NaCl and 0.2% sodium azide. This solution was then incubated with RIAP (10 mg/mL, *w*/*v*) at 37 °C. Ampicillin (0.0005%, *w*/*v*) was added to prevent microbial growth during the reaction. Substrate–enzyme mixtures were collected at intervals of 0, 1, 2, 3, 6, 12, and 24 h, with enzyme activity terminated by heating to 95 °C for 5 min, followed by centrifugation at 7119× *g* for 5 min. The glucose released by enzymatic hydrolysis was quantified using the D-Glucose Assay Kit (Megazyme Co., Bray, Wicklow, Ireland) with maltodextrin and dextran included as controls for comparative analysis.

### 2.8. Statistical Analysis

Data are presented as the mean ± standard deviation for each experiment. Statistical analyses were conducted using SPSS (v27.0; IBM Corp., Armonk, NY, USA) along with one-way analysis of variance and Duncan’s multiple range tests. The results were considered statistically significant at *p* < 0.05.

## 3. Results and Discussion

### 3.1. Growth Properties of G. oxydans

The growth characteristics of the three different pH condition culture groups of *G. oxydans* are shown in Table 1 and Figure 1. The Jp-UC group, in the absence of NaOH-solution feeding, the pH rapidly decreased to 4.0 after 6 h and to 3.0 after 12 h of growth, eventually stabilizing at pH 2.5 for the remainder of the 72-h incubation period. *G. oxydans* oxidizes and metabolizes glycerol into acidic substances in medium [38], resulting in acidification of the medium and subsequent hampering of growth at pH levels <3.5 by completely inhibiting the activities of enzymes in the pentose phosphate pathway [20]. The optimal pH for *G. oxydans* growth lies between 4.0 and 5.0, which promotes the oxidation of acetaldehyde to acetate and shifts the culture medium to lower pH values [39]. After adjusting the medium pH with NaOH, the Jp-4.5 and Jp-5.0 groups maintained stable pH values of 4.5 and 5.0, respectively. These conditions supported increased growth, reflected in OD_600_ values of 7.2 and 6.5 for Jp-4.5 and Jp-5.0, respectively. This resulted in a 1.8- to 1.9-fold increase in cell mass compared to the Jp-UC group, which had an OD_600_ of 3.7. This was in agreement with a previous study that reported that *G. oxydans* showed optimal growth at a pH range of 4.0 to 6.0 [32,40].

Additionally, we observed initial decreases in the dissolved oxygen (DO) content from 6 to 12 h, followed by gradual increases to a plateau after 24 h across all experimental groups. In each case, the cells exhibited an exponential growth phase between 6 and 24 h, likely due to enhanced growth facilitated by the utilization of available oxygen. Moreover, the observed patterns of decreased DO (Jp-4.5 > Jp-5.0 > Jp-UC) and increased growth (Jp-4.5 > Jp-5.0 > Jp-UC) confirmed the expected correlation between DO and growth rate in these obligate aerobic bacteria. Dextran biosynthesis would be possible in cultures grown in maltodextrin-containing media due to the DDase activity of *G. Oxydans* [15]. Given that the enzyme is pH-sensitive, a distinct pattern in dextran production may emerge, potentially leading to differences in the characteristics or composition of the resulting bioconversion products. Yamamoto et al. [15] reported that the optimal and stable pH of DDase ranges from 4.0 to 4.5 and 3.2 to 5.0, respectively.

### 3.2. Physicochemical Properties of Dextran Produced in G. oxydans Culture Media

#### 3.2.1. Flow Behavior

The apparent viscosity values presented in Table 2 revealed distinct differences among the culture media. Jp-4.5 exhibited the highest viscosity (30.99 mPa·s), even surpassing that of a 20% dextran solution (16.17 mPa·s), suggesting significant polymer formation. The Jp-UC medium showed an intermediate viscosity (15.2 mPa·s), comparable to a 10% dextran solution, while Jp-5.0 had a considerably lower viscosity (2.39 mPa·s) that was comparable to the initial medium, implying limited polymer synthesis at pH 5.0.

On the other hand, Figure 2 shows the rheological properties of *Gluconobacter oxydans* culture media under different pH conditions, where the Jp-4.5 medium demonstrated non-Newtonian, shear-thinning behavior, with high shear stress at low shear rates that gradually decreased as the shear rates increased, while the Jp-5.0 and Jp-Uc media exhibited Newtonian fluid behavior. Zarour et al. [41] reported that dextran solutions from five lactic acid bacteria (LAB) sources displayed viscosity behaviors dependent on the concentration and shear rate, with low concentrations maintaining Newtonian, shear rate-independent viscosity similar to certain commercial dextrans, while higher concentrations showed a viscosity decrease with increasing shear rate, indicating non-Newtonian, pseudoplastic behavior. Similarly, Xu et al. [42] found that a high molecular weight dextran solution (5.223 × 10^5^) at 30 wt% also displayed pseudoplastic properties, likely due to hydrodynamic forces disrupting structural entanglements among the α-glucan chains in solution under shear [42]. These suggest that the Jp-4.5 medium likely contains a higher dextran concentration than the Jp-5 and JU media, which may account for its distinctive shear-thinning behavior.

These findings partially align with those of Yamamoto et al. [15] and Sadahiro [43], who reported DDase stability between pH 3.2 and 5.5 and optimal *G. oxydans* growth at pH 5.5. However, under the controlled pH conditions of the jar fermentor used here, pH 5.0 did not support effective dextran synthesis, as reflected in the low viscosity in Jp-5.0. While pH adjustments between 4.5 and 5.0 enhanced *G. oxydans* growth by reducing medium oxidation, our results demonstrate that a narrower pH range, closer to 4.5, is required for optimal dextran production. Thus, pH 4.5 appears to balance both growth and dextran biosynthesis, underscoring the importance of precise pH control to maximize the dextran yield.

#### 3.2.2. Molecular Weight Distribution

Figure 3 and Table 3 show the molecular weight (MW) distribution of dextran synthesized through *G. oxydans* culture, as analyzed by GPC. In Figure 3A, the decrease in substrate (consumption pattern) and the formation of products as well as their size distribution over time are presented. At 0 h (initial medium), the peak eluting at 15–18 min corresponded to the maltodextrin component added as a substrate, with a peak molecular weight (Mp) of 1284 Da, indicating it as a maltooligosaccharide with a degree of polymerization (DP) of 7–8, composed primarily of glucose. It could be observed that the distribution of molecular weights extended to components above 22 kDa. This peak rapidly decreased over the course of 6 and 12 h, suggesting that as the *G. oxydans* culture progressed, lower molecular weight substances were consumed, indicating an efficient utilization of carbon sources and sugar-transferring substrates. At 12 h, the peak at 16 min was significantly lower, indicating that maltodextrin, which had been depleted at the appropriate time, was likely injected as a feeding solution. At 12 h, the supernatant of the culture showed significant differences in peak patterns from the initial medium. As the culture progressed, the peak at 12–14 min corresponding to a 50 kDa molecular weight increased, suggesting effective dextran and polysaccharide production. The molecular weight distribution patterns at 48 and 72 h were similar, with the peaks for low-molecular-weight substances slightly shifting to the left compared to the initial medium, indicating the use of maltodextrin chains as acceptors and the transfer of glucose molecules, leading to an increase in the molecular weight of the products.

In Figure 3B, dextran from the culture supernatants at 72 h under different pH conditions were recovered by ethanol precipitation and compared by preparing 1% (*w*/*v*) aqueous solutions of the dried dextran. The Mp of the Jp-UC sample was significantly smaller than those of Jp-4.5 and Jp-5.0, which were 44,267 Da and 50,722 Da, respectively. This result was consistent with the lower peak observed between 12 and 14 min in the chromatogram. Notably, *G. oxydans* cultured at pH 5.0 showed a decrease in glucose levels without a significant increase in either maltooligosaccharide synthesis or high-molecular-weight substance production (Figure 3B). In evaluating the DDase activity based on substrate specificity, Yamamoto et al. [15] reported that 30.2% and 57.6% of dextran was synthesized from maltohexaose and short-chain amylose, respectively. Additionally, other studies have indicated that dextran synthesized by DDase exhibits a broad molecular weight distribution, ranging from 6.6 kDa to 96.8 kDa [9,24]. A recent study [44] reported conversion into isomalto-megalosaccharides, which differ slightly from dextran, with similar-sized products appearing between 6 kDa and 1.3 kDa in the chromatogram (Figure 3B). Similarly, the polysaccharide composition and yield biotransformed by *G. oxydans*, recovered from media cultured under varying pH conditions, exhibited diverse compositions.

#### 3.2.3. ^1^H-NMR Analysis

Figure 4 shows the ^1^H-NMR spectrum of the dextran isolated from *G. oxydans* culture media, confirming the formation of dextran from dextrin through chemical shifts at 5.3 ppm and 4.9 ppm, which corresponded to the formation of an α-(1→4) and an α-(1→6) linkage (glycosidic bonds in the polymer), respectively [45]. Consistent with our results, Cheetham et al. [46] also observed the anomeric proton resonance for the α-(1→6) D-glycosyl residue at approximately 4.94 ppm in their studies on Dextran T10 and B-512. In our analysis of polysaccharides from each experimental group (Jp-UC, Jp-4.5, and Jp-5.0), the presence of an α-(1→6) linkage was confirmed by a specific peak absent in MD alone (Figure 4). Among the groups, the estimated linkage ratio [α-(1→4) to α-(1→6)] in the polysaccharides from Jp-4.5 was 1:2.84; which was 1.5- and 16-fold higher than that in Jp-UC (1:1.84) and Jp-5.0 (1:0.17), as shown in Table 4. These findings indicate that the DDase activity was affected by the culture pH, with the maximum dextran synthesis observed at pH 4.5 but not at pH 5.0, suggesting that glycosyltransfer activity might be hindered at pH 5.

### 3.3. Digestibility of Dextran by Mammalian α-Glucosidases

To evaluate the digestibility of synthesized dextran into glucose via bioconversion by mammalian α-glucosidase under different pH conditions, the reaction with digestive enzymes derived from rat small intestine tissue was measured for 24 h (Figure 5). α-Glucosidase is an enzyme that breaks down polysaccharides and disaccharides into monosaccharides in the small intestine; its activity and pattern of action are known to vary depending on the evolutionary level of the organism [37,47]. The small intestinal mucosa contains a complex of enzymes, maltase-glucoamylase (MGAM) (EC 3.2.1.20 and 3.2.1.3) and sucrose-isomaltase (SI) (EC 3.2.148 and 3.2.10), which have different actions to digest carbohydrates and hydrolyze them to free glucose [48]. Mammalian digestive enzymes derived from rat intestine were used for these measurements instead of AMG, a microbial enzyme commonly utilized in digestion studies, because they closely mimic digestion in the human body [49].

We found that 84.8% of the maltodextrin used as a substrate was hydrolyzed to glucose within the first hour of digestion compared to 49.6% and 59.1% for Jp-UC and Jp-5.0, respectively, and only 32.8% for Jp-4.5. Over a total of 24 h, Jp-UC released 82.3% of glucose, showing a similar level of hydrolysis to that of maltodextrin (90.6%). In contrast, Jp-4.5 exhibited a total glucose release of only 54.0%, indicating a clear pattern of slow glucose decomposition and highlighting significant differences in dextran digestibility depending on the pH conditions during culture. This finding aligns with the glycosidic linkage ratios shown in Figure 4; maintaining the pH at 4.5 during *G. oxydans* fermentation led to active dextrin-to-dextran converting enzyme (DDase) activity, transforming the substrate into dextran, an α-1,6 glucan. Meanwhile, commercial dextran derived from sucrose, used as a positive control for the digestion reaction, showed an initial glucose release of 5.7%, increasing to 33.6% by the end of the 24 h reaction.

Maltodextrin, an α-1,4 glucan, is hydrolyzed by amylase in the body to produce maltose units. These maltose units are either directly absorbed by the intestinal epithelium or further broken down by maltase to release glucose, resulting in a rapid rise in blood glucose levels [50,51]. However, studies have shown that α-1,6 bonds are hydrolyzed more slowly than α-1,4 bonds, which reduces in vitro digestibility [48]. Thus, increasing the number of α-1,6 bonds while decreasing α-1,4 bonds is significantly associated with improved slow digestibility, and recent research has focused on increasing the proportion of α-1,6 bonds [47,52]. This reduction in the rate of initial glucose breakdown is crucial as it helps prevent blood glucose spikes, thereby suppressing insulin secretion and potentially lowering the metabolic load on organs involved in digestion [53]. Thus, cultured solutions synthesized through the *G. oxydans* enzyme reaction and their product by ethanol precipitated under specific pH conditions may serve as functional ingredients to prevent blood sugar spikes and regulate postprandial glycemic response by promoting slow digestion.

## 4. Conclusions

In conclusion, this study demonstrated that appropriate pH adjustments are critical in optimizing *G. oxydans* growth and dextran production, with pH 4.5 proving particularly effective compared to uncontrolled pH and pH 5.0. The highest apparent viscosity (30.99 mPa∙s) and α-1,6 linkage ratio observed at pH 4.5 indicated enhanced enzyme activity and efficient dextran synthesis. Molecular size distribution analysis confirmed that dextran molecular weights ranged from 1.3 × 10^3^ Da to 5.1 × 10^4^ Da, with a broader size range under the tested pH conditions. Moreover, the dextran produced under the Jp-4.5 condition exhibited the slowest glucose release, suggesting the production of α-1,6 glucan with slow-digesting characteristics. Therefore, dextran bio-converted from dextrin by *G. oxydans* has potential as a functional carbohydrate ingredient to help manage postprandial glycemic response.

## Figures and Tables

**Figure 1 foods-14-00085-f001:**
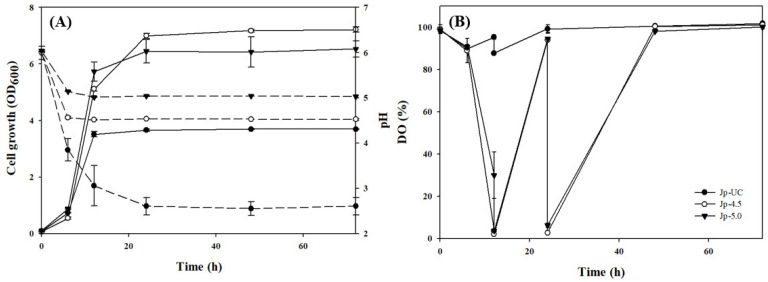
Changes in the *G. oxydans* growth curve (solid line), and pH (short dash line) (**A**); DO (**B**). Symbols: ●, Jp-UC; ○, Jp-4.5; and ▼, Jp-5.0.

**Figure 2 foods-14-00085-f002:**
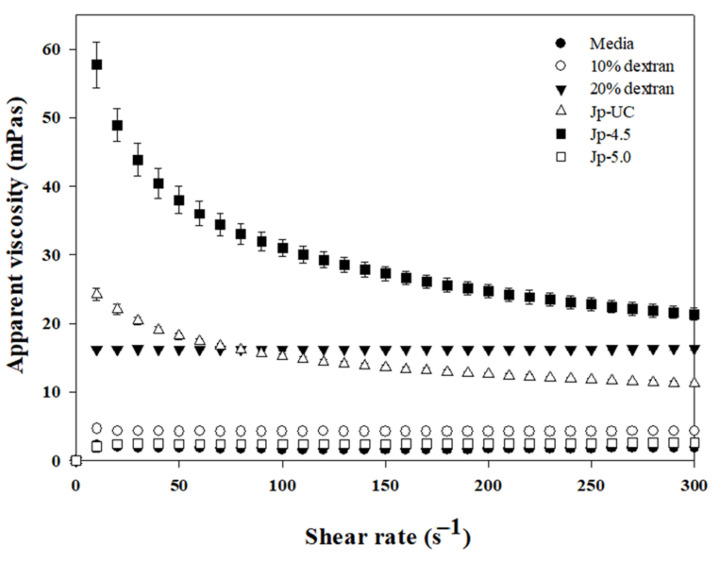
Effect of shear stress on *G. oxydans* culture media growth under various culture conditions.

**Figure 3 foods-14-00085-f003:**
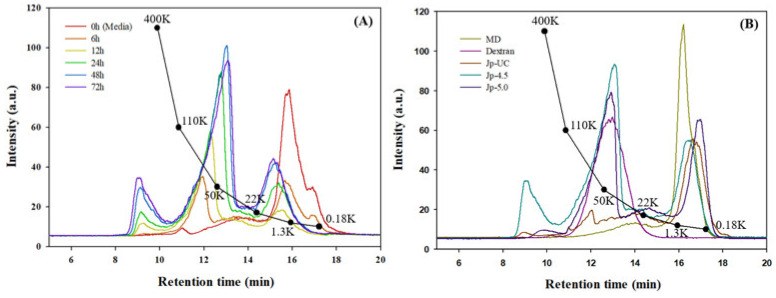
Gel permeation chromatography analysis of dextran from *G. oxydans*. (**A**) Time-dependent changes in the molecular weight distribution of Jp-4.5. (**B**) Molecular weight distribution of dextran produced under varying pH conditions.

**Figure 4 foods-14-00085-f004:**
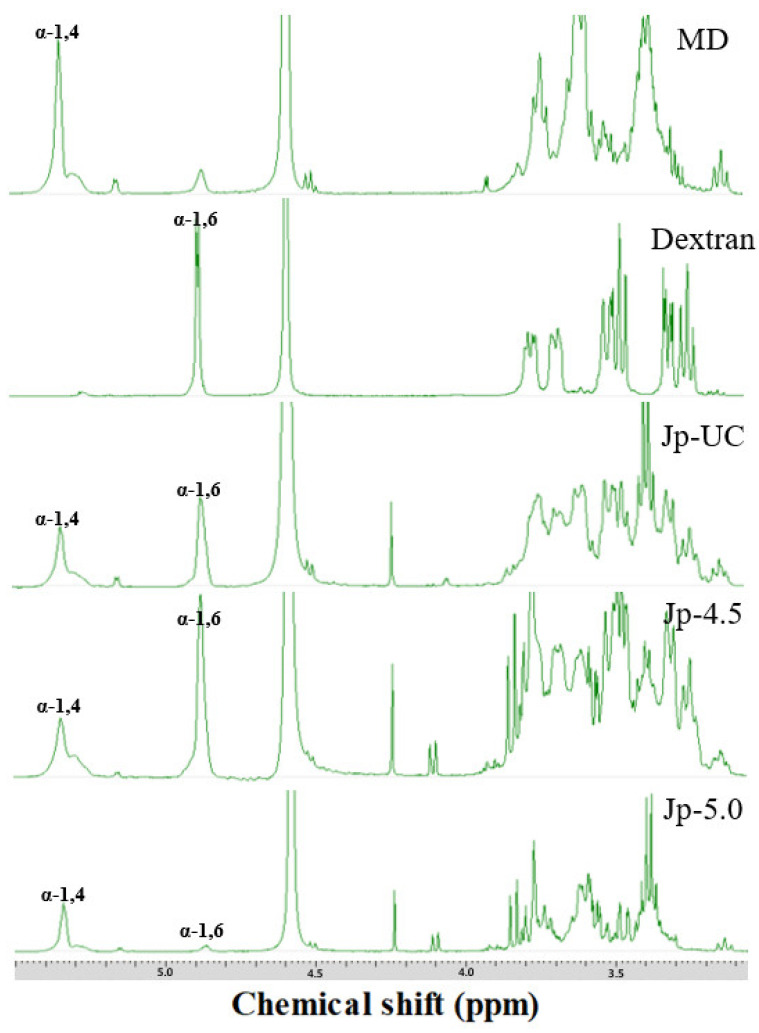
^1^H-NMR spectrum of dextran from *G. oxydans*.

**Figure 5 foods-14-00085-f005:**
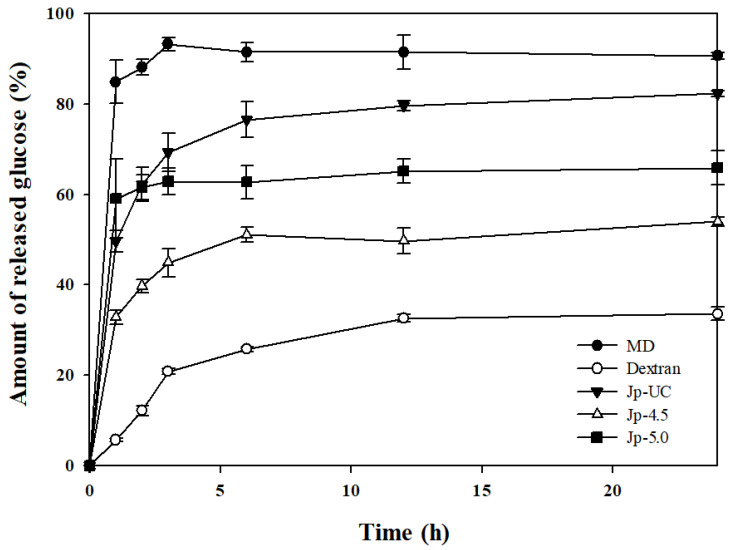
Glucose generation property of dextran from *G. oxydans* by mammalian small intestinal α-glucosidase.

**Table 1 foods-14-00085-t001:** Jar fermenter culture design with different pH conditions.

Sample Abbreviation	Composition		Condition			
	* Substrate Solution (g)	Media Volume (L)	RPM	pH	Time (h)	Aeration (vvm)
Jp-UC	120	1.2	600	Unadjusted	72	0.5
Jp-4.5				4.5		
Jp-5.0				5.0		

* Total substrate concentration with feeding at 12 h and 24 h. Jp-UC: Jar fermentor pH uncontrolled. Jp-4.5: Jar fermentor pH 4.5. * Jp-5.0: Jar fermentor pH 5.0. RPM, revolutions per min.

**Table 2 foods-14-00085-t002:** Rheological properties of the culture media.

Sample	Apparent Viscosity (mPa∙s)
Media	1.67 ± 0.028 ^d^
10% dextran solution	4.25 ± 0.059 ^c^
20% dextran solution	16.17 ± 0.123 ^b^
Jp-UC	15.20 ± 0.407 ^b^
Jp-4.5	30.99 ± 1.259 ^a^
Jp-5.0	2.39 ± 0.349 ^d^

Data in the same column with different letter represent a significant difference at *p* < 0.05.

**Table 3 foods-14-00085-t003:** MW distributions of samples.

Sample	MW ^(1)^	Mp ^(2)^	DP ^(3)^
MD	5636	1284	8
Dextran	41,559	34,780	215
Jp-UC	28,669	1512	9
Jp-4.5	88,286	44,267	273
Jp-5.0	46,577	50,722	313

^(1)^ Mw, total weight average molecular weight (kDa); ^(2)^ Mp, molecular weight of highest peak (kDa); ^(3)^ DP, degree of polymerization, expressed by dividing the Mp value by the molecular weight of one glucose molecule (162 Da).

**Table 4 foods-14-00085-t004:** Analysis of the ratio of glycosidic linkages in polysaccharides derived from *G. oxydans*.

Samples	Glycosidic Linkage Ratio	
	α-(1→4)	α-(1→6)
Jp-UC	1.00	1.84 ± 0 ^b^
Jp-4.5	1.00	2.84 ± 0.02 ^a^
Jp-5.0	1.00	0.17 ± 0.01 ^c^

Data in the same column with different superscript represent a significant difference at *p* < 0.05

## Data Availability

The original contributions presented in this study are included in the article/Appendix A. Further inquiries can be directed to the corresponding author.

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
