# Peer review of "Synthesis and Physico-Chemical Analysis of Dextran from Maltodextrin via pH Controlled Fermentation by Gluconobacter oxydans"

_foods, 2025, doi:10.3390/foods14010085_

Round 1

Reviewer 1 Report

Comments and Suggestions for Authors

This study analyzed the role of pH in jar fermentor on the growth of G. oxydans and the synthesis of dextran. Since fermentation was conducted in a complex environment, including many influencing factors (temperature, pH, rotating and media composition). Therefore, fermentation for production of dextran should be considered in a comprehensive view, instead of considering pH only. Therefore, this manuscript should be rejected.

Comments on the Quality of English Language

Language should be improved.

Author Response

Comment 1: This study analyzed the role of pH in jar fermentor on the growth of G. oxydans and the synthesis of dextran. Since fermentation was conducted in a complex environment, including many influencing factors (temperature, pH, rotating and media composition). Therefore, fermentation for production of dextran should be considered in a comprehensive view, instead of considering pH only. Therefore, this manuscript should be rejected.

Response 1: Thank you for your feedback. We acknowledge that various factors influence the fermentation process of microorganisms. However, this study specifically focuses on the activity of the dextransucrase (DDase) enzyme, which occurs concurrently with the cultivation of G. oxydans.

To address this, the experiment was designed to prioritize pH as the primary factor affecting enzyme activity, given that pH and temperature are critical variables in this context. Among these factors, pH showed the most significant impact on enzymatic activity. This is evident in the Jp-UC experimental group, where pH was not controlled as an independent variable. In this group, the linkage structure of the product (dextran) resulting from enzymatic activity remained largely unchanged, and the viscosity change was minimal. This indicates a lower production of polymeric dextran under these conditions.

By focusing on pH, this study provides valuable insights into its role in dextran synthesis, while acknowledging that other factors also contribute to the overall fermentation process.

Reviewer 2 Report

Comments and Suggestions for Authors

The study with the title Synthesis and Physico-chemical Analysis of Dextran from Maltodextrin via pH controlled Fermentation by Gluconobacter oxydans analyzed the influence of pH on the fermentation process and dextran synthesis under different conditions (without pH control, pH 4.5 and pH 5.0, Jp-UC, Jp-4.5, and Jp-5.0). Dextran is an exopolysaccharide (EPS) with various uses in the food and pharmaceutical fields, being synthesized mainly by Leuconostoc mesenteroides. It can also be produced from dextrin by fermentation carried out by Gluconobacter oxydans, using dextran dextrinase. After 72 hours of fermentation, the cell density was maximum at pH 4.5, and viscosity analysis indicated the highest EPS production under these conditions. The structure of dextran was confirmed by 1H-NMR, revealing the presence of α-1,6 glycosidic bonds, and the molecular weight varied depending on pH between 1.3 × 10³ Da and 5.1 × 10⁴ Da. The rate of hydrolysis to glucose, which reflects the digestibility of carbohydrates, was lowest at pH 4.5. The conclusions suggest that adjusting pH can optimize the production of dextran, which could be used as a slowly digested carbohydrate, contributing to the regulation of postprandial blood glucose levels and having potential benefits in the management of chronic metabolic diseases such as diabetes.

Author Response

We sincerely appreciate the reviewer’s thoughtful summary of our study and acknowledgment of its significance. We are pleased that the key findings, including the influence of pH on dextran synthesis, its molecular characteristics, and potential health applications, have been well understood.

Regarding the production of dextran as a slowly digestible carbohydrate, we agree that this could have substantial benefits in managing chronic metabolic conditions. Our study aimed to provide fundamental insights into optimizing dextran production, and we are optimistic about its applicability in food and pharmaceutical industries, particularly for the development of functional foods targeting blood glucose regulation.

Reviewer 3 Report

Comments and Suggestions for Authors

Dear editor:

Thank you for the invitation to review this manuscript. It is an interesting article on the synthesis of dextran from maltodextrin with important repercussions in antianemic treatments. 

The introduction is clear, where the authors highlight the background of the work and the objective is defined. 

Materials and methods: indicate the references in the methodologies described in case they are not from the authors, since there are no references in the methodologies. It is also important to include a flow chart to make the process carried out clearer. 

In the results and discussion, at the beginning it is not clear why we acidified but the authors say that in the absence of NaOH. What happens to the pH of G. oxydans when alkalis is added? The optimal pH (4-5) must be supported with bibliography since there is a lot of bibliography and it only says 1. 

What relationship exists between dissolved oxygen and growth? 

Is there a relationship between Newtonian velocity and pH? At what concentration of dextran does the fluid behave as non-Newtonian?

In NMR analyses indicate the coupling constants and if you also have 2D NMR data to support your results.

Author Response

Dear Reviewer

Thank you for your thoughtful feedback on our manuscript. We appreciate your suggestions for improving clarity and consistency in the presentation of our work.

Comments:The introduction is clear, where the authors highlight the background of the work and the objective is defined. 

  • Materials and methods: indicate the references in the methodologies described in case they are not from the authors, since there are no references in the methodologies. It is also important to include a flow chart to make the process carried out clearer. 

Response: Thank you for this valuable suggestion. In response to your comment: We have carefully reviewed the methodologies section and added references for any methods adapted from previous studies to ensure proper attribution (line 116, 131, 146). To improve clarity, we have included a flowchart (As a supplementary figure) that visually outlines the experimental process, detailing key steps that could help readers better understand the workflow of our study.

  • In the results and discussion, at the beginning it is not clear why we acidified but the authors say that in the absence of NaOH. What happens to the pH of G. oxydans when alkalis is added? The optimal pH (4-5) must be supported with bibliography since there is a lot of bibliography and it only says 1. 

Thank you for your insightful comment. As G. oxydans grows, it oxidizes glycerol in the medium, producing acidic metabolites that gradually lower the pH of the medium. To prevent complete acidification and maintain suitable conditions for dextransucrase (DDase) activity, an alkali such as NaOH is added to control the pH. In the JP-UC experimental group, where NaOH was not added, the medium became completely acidified as fermentation progressed. This acidification hindered enzymatic activity and consequently reduced dextran production. We have clarified this explanation in the revised manuscript (Line 176) for improved understanding. Regarding the optimal pH range (4–5) for G. oxydans, additional references have been included to support this claim. Furthermore, we have incorporated references highlighting the pH conditions required for DDase activity to provide a more comprehensive discussion, as you suggested.

  •  
  • What relationship exists between dissolved oxygen and growth? 

Response: Thank you for this question. Obligate aerobic bacteria, such as G. oxydans, rely on oxygen for their metabolic processes and growth. As a result, a reduction in dissolved oxygen content in the medium can serve as a potential indicator of microbial activity and growth. In our study, the decrease in dissolved oxygen observed between 6 and 12 hours in Fig. 1(B) corresponds to the exponential growth phase of the cells, which spans approximately from 6 to 24 hours. This relationship highlights the critical role of dissolved oxygen in supporting the metabolic activities and proliferation of G. oxydans.

  • Is there a relationship between Newtonian velocity and pH? At what concentration of dextran does the fluid behave as non-Newtonian?

Response: Thank you for your thoughtful question. While there is no direct relationship between Newtonian viscosity and pH in a mechanistic sense, pH can indirectly affect viscosity by influencing the molecular weight (Mw) of dextran produced during cultivation. For instance, in media that became acidic without pH control, the activity of dextransucrase (DDase) was reduced, resulting in lower production of high molecular weight substances like dextran. Consequently, the viscosity of the cultivation medium was lower under these conditions.

Viscosity and the transition from Newtonian to non-Newtonian behavior are primarily governed by the combination of Mw and concentration of dextran, which together determine the fluid's behavior under shear stress. As noted in Section 3.2.2, the dextran produced from jar fermentation exhibited a higher molecular weight compared to the commercial dextran (40 kDa) used in our study. This higher Mw led to non-Newtonian behavior due to increased molecular chain entanglements and hydrogen bonding, particularly at higher concentrations.

There is no specific concentration at which dextran universally exhibits non-Newtonian behavior. Instead, it is the combination of Mw and concentration that dictates this transition. For example: At low Mw, such as the commercial dextran (40 kDa) used in our study, the solutions behaved as Newtonian fluids even at concentrations up to 50%. Conversely, Xu et al. (2008) reported that dextrans with higher Mw (e.g., 522 kDa) exhibited pseudoplastic properties at 30 wt%, while dextran with an intermediate Mw of 109 kDa also showed non-Newtonian behavior at the same concentration. The general trend is that as Mw increases, physical chain entanglements and inter- and intramolecular hydrogen bonds become more significant. At low shear rates, these interactions dominate, resulting in higher viscosity. As shear rate increases, the molecular chains disentangle, and hydrogen bonds break, leading to shear-thinning behavior.

  • In NMR analyses indicate the coupling constants and if you also have 2D NMR data to support your results.

Response: Thank you for your insightful comment to improve our manuscript. In this study, we utilized ¹H-NMR to analyze the anomeric carbon region, specifically focusing on the glucose linkages to distinguish between α-1,4 and α-1,6 glycosidic bonds. This approach was critical for confirming whether the α-1,4 linkages in starch-derived maltodextrin were successfully converted into α-1,6 linkages through transglycosylation.

Regards!

Reviewer 4 Report

Comments and Suggestions for Authors

The paper investigated how the pH of culture media affect the synthesis and properties of dextran produced by Gluconobacter oxydans from MD, including its digestiblity by mammalian α-glucosidase, which benefited  developing a functional carbohydrate ingredient to manage postprandial glycemic response by bio-conversion way. Although the investigations and results are reasonable, the quality of this article could be improved if these inquiries are addressed: 1) In introduction and result analysis, because the Gluconobacter oxydans used can also produce bacterial cellulose (BC), the authors should introduce this and compare the different culture conditions of yielding dextran and BC. 2) DDase activities in the cells should be determined for all culture conditions to well explained the results obtained. The other comments and suggestions for revisions are listed bellows.

1. What unit of cell density? (Line 21-22) 

2. Line 43,45, 228, the researchers' surname should be mentioned and pay attention to correct use of restrictive and non-restrictive relative clauses.

3. Line 84, why the pH range (6.6-7.0) used for preparation of stock culture was so different from those for jar fermenter cultivation (UC, 4.5 and 5.0)? 

4. Line 147, The conditions of 1H-NMR analysis should be in detail.

5. Line 149, culture medium? EPS.  The authors should also distinguish the EPS and dextram in result report.

6. Line 182, in agrement should be in agreement.

7. Line 239, raw should be corrected into column.

8. For Figs. 3, 4, 5 and Tables 3,4, the samples should be made clear in annotation. The Dextran used is from commerical? 

9. Line 297, showen should be shown.

10. Line 351, the Jp-4.5 condition should be corrected as the dextram produced under Jp-4.5 condition.

Author Response

Dear Reviewer,

We would like to express our sincere gratitude for your insightful comments and valuable suggestions regarding our manuscript. Your thorough review has provided invaluable guidance, and we appreciate the opportunity to address your questions and clarify certain aspects to enhance the quality of our work.

Reviewer 4 Comments and Suggestions for Authors

The paper investigated how the pH of culture media affect the synthesis and properties of dextran produced by Gluconobacter oxydans from MD, including its digestiblity by mammalian α-glucosidase, which benefited developing a functional carbohydrate ingredient to manage postprandial glycemic response by bio-conversion way. Although the investigations and results are reasonable, the quality of this article could be improved if these inquiries are addressed:

  1. In introduction and result analysis, because the Gluconobacter oxydans used can also produce bacterial cellulose (BC), the authors should introduce this and compare the different culture conditions of yielding dextran and BC.

Response: Thank you for your thoughtful comment. We have reviewed research findings on the production of bacterial cellulose (BC) by Gluconobacter oxydans. These studies indicate that BC synthesis occurs when the pH drops below 4 during the proliferation of G. oxydans. In contrast, our findings demonstrate that maximum dextran production occurs at a pH of 4.5 under controlled conditions.

Additionally, the differences between BC and dextran—outlined in the revised introduction as – “Acetic acid bacteria (AAB), such as Gluconobacter oxydans, can produce β-glucan structures like bacterial cellulose (BC) or ⍺-glucans structures like dextran under specific conditions [6, 7]. BC is a water-insoluble polymer composed of β-1,4 glucan chains, which differentiate it from dextran, a water-soluble polymer primarily composed of α-1,6 glucosidic linkages with variable amounts of α-1,4-, α-1,3-, and α-1,2 linkages [6, 8, 9, 10]. The production of these biopolymers depends on culture conditions, such as pH, medium composition, and enzymatic activity”, highlight their distinct structural properties, production conditions, and isolation methods.  Given these distinctions, our study focuses exclusively on dextran production under conditions optimized to favor its synthesis. This approach aligns with the specific goals of our research.

  1. DDase activities in the cells should be determined for all culture conditions to well explained the results obtained. The other comments and suggestions for revisions are listed bellows.

Response: Thank you for this valuable feedback. Dextransucrase (DDase) activity has been extensively documented to utilize maltodextrin as a substrate for transglycosylation, leading to the production of dextran or isomaltomegalosaccharides (citation + additional reference). In our study, we used the same Gluconobacter oxydans strain previously reported for its DDase activity. The effective production of dextran and the degree of glycosylation, including the formation of α-1,6 linkages, were confirmed through product analysis and evaluated using ¹H-NMR.

Based on insights from prior studies and feedback from the research team that developed the method, expressing DDase through a heterologous system has been noted to pose significant challenges. For this reason, our study employed a whole-cell reaction (fermentation) approach using Gluconobacter oxydans. This method allowed us to examine the enzyme’s activity in situ under various pH-controlled culture conditions. We believe this approach adequately reflects the enzymatic activity in different conditions without isolating DDase. However, we acknowledge that future studies could include direct measurement of DDase activity to further substantiate these findings.

  1. What unit of cell density? (Line 21-22) 

Response: Thank you for your question. Among the various methods available for measuring microbial growth, we employed a spectrophotometric approach to determine cell density. Absorbance was measured at 600 nm (OD600). Since absorbance is a dimensionless value derived from the optical density of the culture, no specific units are provided.

  1. Line 43,45, 228, the researchers' surname should be mentioned and pay attention to correct use of restrictive and non-restrictive relative clauses.

Response: Thank you for your suggestion. We have revised the manuscript by adding the researchers' surnames before the references, as recommended (Line 43,46). Additionally, we have ensured the correct usage of restrictive and non-restrictive relative clauses throughout the manuscript for improved clarity and accuracy.

  1. Line 84, why the pH range (6.6-7.0) used for preparation of stock culture was so different from those for jar fermenter cultivation (UC, 4.5 and 5.0)? 

Response: The initial pH of the prepared medium ranges from 6.6 to 7.0 (intrinsic property of the media). However, as G. oxydans grows, it produces acidic substances (line 172), causing the pH of the medium to decrease over time. Since uncontrolled pH negatively impacts the growth of G. oxydans and the reaction of DDase activity, the pH was adjusted to 4.5 and 5.0 by adding NaOH.

  1. Line 147, The conditions of 1H-NMR analysis should be in detail.

Response:Thank you for your suggestion. We added the specific condition of 1H-NMR parameter method in the manuscript (line 147 – 151: frequency = 500.16 MHz; spectral width = 9384.4 Hz; acquisition time = 3.5 s; relaxation delay = 20 s; number of scans = 16; 90° pulse width = 7.6 μs; pulse angle = 45°; digital resolution = 0.29 Hz.

  1. Line 149, culture medium? EPS.  The authors should also distinguish the EPS and dextram in result report.

Response: Thank you for your comment. In Line 149, we have replaced the term “EPS” with “Dextran,” as the primary focus of this study is on the production of dextran. Using “Dextran” is more appropriate and specific to the objective of our research. Additionally, we have revised all “EPS” in the material and methos as well as in result and discussion with “dextran” to clarify the distinction between EPS (extracellular polysaccharides) and dextran. Dextran is a specific type of EPS characterized by its α-1,6 glycosidic linkages, which we confirmed through H-NMR analysis.

  1. Line 182, in agrement should be in agreement.

Response:Thank you for pointing out this typing error. We have corrected "in agrement" to "in agreement" in the revised manuscript.

  1. Line 239, raw should be corrected into column.

Response: Thank you, We have corrected "raw" to "column" in the revised manuscript (line 244).

  1. For Figs. 3, 4, 5 and Tables 3,4, the samples should be made clear in annotation. The Dextran used is from commerical? 

Response:Thank you for your suggestion. Annotations were added to all tables and figures. We used commercial dextran as a positive control in all experiments.

  1. Line 297, showen should be shown.

Response:Thank you for bringing this to our attention. We have corrected "showen" to "shown" in the revised manuscript.

  1. Line 351, the Jp-4.5 condition should be corrected as the dextram produced under Jp-4.5 condition.

Response:Thank you for your suggestion. We have revised the text to correctly state "the dextran produced under Jp-4.5 condition" in the updated manuscript.

Regards!

Round 2

Reviewer 1 Report

Comments and Suggestions for Authors

Well answered. Accept